# Task-Shifting: Can Community Health Workers Be Part of the Solution to an Inactive Nation?

**DOI:** 10.3390/ijerph20176675

**Published:** 2023-08-29

**Authors:** Estelle D. Watson, Shabir Moosa, Dina C. Janse Van Rensburg, Martin Schwellnus, Estelle V. Lambert, Mark Stoutenberg

**Affiliations:** 1Department of Exercise Science and Sports Medicine, School of Therapeutic Sciences, Faculty of Health Sciences, University of Witwatersrand, Johannesburg 2193, South Africa; 2Department of Exercise Science, Faculty of Science, University of Auckland, Auckland 1023, New Zealand; 3Department of Family Medicine and Primary Care, Faculty of Health Sciences, University of Witwatersrand, Johannesburg 2193, South Africa; 4Section Sports Medicine, Faculty of Health Sciences, University of Pretoria, Pretoria 0028, South Africa; 5UCT Research Centre for Health through Physical Activity, Lifestyle and Sport (HPALS), Department of Human Biology, Division of Research Unit for Exercise Science and Sports Medicine, Division of Physiological Sciences, Faculty of Health Sciences, University of Cape Town, Cape Town, 7700, South Africa; 6College of Public Health, Temple University, Philadelphia, PA 19122, USA

**Keywords:** advocacy, behavior change, community health workers, health promotion, physical activity

## Abstract

**Background:** In low-to-middle income countries (LMICs), there is a growing burden of non-communicable diseases (NCDs) placing strain on the facilities and human resources of healthcare systems. Prevention strategies that include lifestyle behavior counseling have become increasingly important. We propose a potential solution to the growing burden of NCDs through an expansion of the role for community health workers (CHWs) in prescribing and promoting physical activity in public health settings. This discussion paper provides a theoretical model for task-shifting of assessment, screening, counseling, and prescription of physical activity to CHWs. Five proposed tasks are presented within a larger model of service delivery and provide a platform for a structured, standardized, physical activity prevention strategy aimed at NCDs using CHWs as an integral part of reducing the burden of NCDs in LMICs. However, for effective implementation as part of national NCD plans, it is essential that CHWs received standardized, ongoing training and supervision on physical activity and other lifestyle behaviors to optimally impact community health in low resource settings.

## 1. Introduction

### 1.1. The Burden of Non-Communicable Diseases

In South Africa (SA), non-communicable diseases (NCDs) are emerging as one of the major burdens facing the country’s healthcare system. While much of the primary healthcare system’s resources are aimed at treating infectious and communicable diseases, research shows that NCDs have overtaken HIV/AIDS as the leading cause of death in some areas [1,2,3]. Issues of globalization, inequality, and urbanization all contribute toward the growing prevalence of NCDs and their risk factors [4]. SA has gone through drastic economic change and urbanization in the last 20 years since apartheid. This has magnified the trends in NCD-associated risk factors, such as unhealthy diets and physical inactivity, with NCDs currently accounting for 21% of life years lost [5,6] and resulting in an estimated cost of INT$120.4 billion in SA in 2019 [7].

### 1.2. Physical Activity as a “Medicine” for NCDs

Physical inactivity has been equated to smoking and obesity in its contribution to health risk [8]. Although there is little doubt that being active is associated with a lower mortality risk [9], many people still do not meet the recommended guidelines of 150 min of moderate-to-vigorous intensity physical activity per week [10]. In SA, prevalence data show that an estimated 46% of the population is insufficiently active [5,11], with much of this burden being carried in lower socioeconomic communities [12]. Globally, low physical activity levels are associated with an estimated 9% of premature mortality and up to 10% of all NCDs [8]. In SA, physical inactivity was estimated to contribute to 14% of all-cause mortality, the third highest rate in Africa and greater than many high-income countries, such as Canada (9.1%) and the US (10.8%) [8]. This burden comes with a cost that many low-to-middle income countries (LMICs), like SA, can ill afford. Ding et al. [13] conservatively estimated that physical inactivity costs INT$67.5 billion annually through health care expenses and losses in productivity. In SA, physical inactivity-related costs were the highest for any African country, amounting to 0.69% of total health care expenditures. This economic burden was equal to other high-income countries, such as Canada (0.57%) and the United Kingdom (0.87%). Thus, the overall burden of physical inactivity in SA is above both global and African averages [8,14], making the promotion of physical activity for the prevention and management of NCDs an urgent national priority.

### 1.3. The Role of the Community Healthcare Worker

In LMICs, the dual burden of communicable and NCDs has increased the strain on the health care system’s facilities and human resources, specifically at the primary health care level. A growing solution to this human resource and service provision crisis is a greater inclusion of the community health care worker (CHW) in primary and secondary prevention efforts [15,16]. CHWs typically have little to no formal education, but are provided varying levels of formal and/or in-service training [17]. CHWs may contribute to patient management at a community level and complement specialized health care services [18]. The 2010 U.S. National Community Health Worker Advocacy Survey outlined seven core roles for CHWs [19], which included the following: (1) bridging the gap between communities and health care services, (2) providing culturally appropriate and accessible health education and information, (3) assuring that communities obtain the services they need, (4) providing informal counseling and social support, (5) advocacy for individual and community needs, (6) provision of direct services, and (7) community empowerment and capacity building [16,18,19]. 

CHWs are usually selected by the community, stakeholders, or local organizations to provide basic health care, such as screening, medication management, and health promotion [20], and they work on a continuum between an informal worker and para-professional role [21]. CHWs are often embedded in their community, which provides them with a unique understanding of the culture, language, health concerns, and needs of the community [22,23]. CHWs serve as an affordable first level contact within a primary health care system and play an important role in disease management and onward medical referral in traditionally disadvantaged communities [22,24]. In countries such as SA, where there is a shortage of health professionals, the role of the CHW is much more multi-dimensional [25]. CHWs work across many different settings, such as local health clinics, community events, and are often utilized to conduct home visits. A recent study by Stoutenberg et al. [26] demonstrated that CHW-led home visits to conduct health screenings in a low resource setting in SA were highly feasible (e.g., reaching a large number of households in a short period of time) and well accepted (>95% of households agreed to participate).

There is promising evidence that these CHW-led interventions improve health outcomes. In SA, postpartum mothers who received home visits from CHWs were more likely to attend antenatal care, exclusively breastfeed [27], and report fewer depressive symptoms at follow up [28]. Other studies involving CHWs have demonstrated improvements in patient self-management behaviors, health care utilization, and antihypertensive medication adherence [29,30]. A review conducted by Norris et al. [31] showed that CHW-led interventions were effective in improving patient satisfaction and increasing knowledge levels, as well as significantly reducing healthcare utilization (emergency attendance, hospital admissions) and the associated cost benefits. Similarly, a US home-based CHW study showed an annual cost saving of up to $2245 per patient through reductions in emergency department visits and hospital admissions [25].

### 1.4. Task-Shifting

In many LMICs, the demand for health care exceeds the supply of health workforce to provide it [32]. In fact, it is anticipated that by 2050, there will be insufficient professional health providers to meet the need for care [27]. In South Africa, availability of and access to physicians can be a challenge, particularly in rural or previously disadvantaged areas. In response to this health workforce shortage, task-shifting, a “process of delegation whereby tasks are moved, where appropriate, to less specialized health workers”, Ref. [23] has already been implemented informally in many countries. Coupled with task-sharing (where certain tasks are shared among different healthcare workers), task-shifting can help provide better access to care by using the available human resources more effectively [33]. Several systematic reviews have shown that task-shifting, using a range of non-physician health workers, may improve health outcomes and can potentially be an effective and affordable strategy for the prevention and management of NCDs [33,34,35,36]. Similarly, a meta-analysis by Maria et al. [35] found task-sharing to be moderately effective in managing diabetes in combination with nurses or pharmacists while Anand et al. [37] found that task-sharing with CHWs resulted in a reduction in systolic blood pressure by nearly 4 mmHg. The challenge in many countries remains integrating CHWs and task-shifting as a seamless part of the primary health care (PHC) system. Effective task-shifting involving CHWs often requires health system restructuring and reorganization [23] and should be guided by formal frameworks and available tools [38,39]. The WHO provides several recommendations for task-shifting, including providing clearly defined roles, adequate and ongoing training, as well as evaluation and supervision [23,37,38,40].

### 1.5. Evidence for CHWs as an Integral Part of Healthcare

In SA, the role of CHWs as an integrated part of SA health care was formally recognized in the re-engineering of the SA primary health care (PHC) system launched by the National Department of Health in 2011 [41]. The vision of the re-engineering was to support greater preventive and health-promoting community-based care by deploying CHWs as the cornerstone of municipal ward-based PHC outreach teams (also known as WBPHCOTs) [42]. Under the supervision of professional nurse supervisors, CHWs were envisaged to conduct activities such as assessing community health needs, providing health education, promoting healthy behaviors (e.g., physical activity), and serving as a bridge connecting patients and communities to health services and local health clinics. Approximately 70,000 CHWs (or health professionals in similar roles) were initially integrated into communities across SA as part of the WBPHCOTs [43].

During the first decade of PHC re-engineering, numerous barriers were encountered in the initial rollout of the WBPHCOTs, such as supervision issues (e.g., shortage of professional nurses), working conditions, and inadequate CHW preparation [44]. However, there remains great optimism that the re-engineering of PHC is a necessary step in re-orienting the SA health system to a people-centered approach with a focus on promoting health and preventing/managing disease. Numerous WBPCHOTs were established in areas of greatest need to improve health services, yet there remains considerable potential for WBPHCOTs to focus on addressing social determinants of health related to NCDs (e.g., food environments, pedestrian safety, access to health services) [45]. A national evaluation conducted in 2019 suggests that the WBPHCOTs are a necessary and critical program that should be continued and strengthened [46].

## 2. Community Health Workers and NCD Prevention

### 2.1. The Growing Role of CHWs in NCD Prevention and Treatment

There is an established body of literature describing the effectiveness of CHWs and their role in NCD prevention and control, particularly in higher income countries. In the U.S., randomized control trials demonstrate that CHW-led interventions result in significant improvements in blood pressure, glycemic control, lowering of HbA1c concentration, and diabetes medication utilization [47,48,49]. Contrary to the U.S., CHWs in Sub-Saharan Africa have largely focused on communicable diseases, such as HIV/AIDS, tuberculosis, maternal and infant health and, more recently, responding to the COVID-19 pandemic [50]. In a review by Schenider et al. [51] of work published between 2005 and 2014, there was nearly a seven-fold increase in publications involving CHWs (half from Africa), yet only 4.5% of these articles focused on NCD prevention and treatment. However, numerous calls to action have advocated for greater inclusion of CHWs in NCD prevention and treatment in LMICs [52,53,54]. Tsolekile et al. [23] proposed that CHWs can play a pivotal role in management of NCDs through assessing risk, supporting medication adherence and education on self-management. Further, a systematic review by Joshi et al. [33] found that involving CHWs in a task-shifting model of care is a viable, clinically effective, and cost-effective strategy for improving access to healthcare for NCDs. In SA, CHWs not only understand the health concerns of the communities they serve, but often share common health risk profiles and, at times, ambivalence concerning lifestyle changes, including obesity and physical inactivity experienced by those underserved communities [55]. Despite these potential barriers, their role in NCD prevention has gradually emerged in response to the unique needs of the communities they serve [16,19].

### 2.2. NCD Treatment and Control

Multiple studies have examined the role of CHWs in the self-management of NCDs, such hypertension, heart disease, and stroke [29,56]. In a review of RCTs, all but one study showed significant improvements in blood pressure control with CHW interventions [56]. Similarly, another systematic review of thirteen randomized controlled trials assessing the efficacy of CHW interventions to lower HbA1c showed modest reductions over a 12-month period when compared to usual care, and this effect was more pronounced when targeting at-risk patients [57]. In addition to demonstrating effectiveness, CHW-led interventions for NCD treatment and control have also been shown to be cost-effective [58,59]. 

### 2.3. Preventing NCDs

In addition to assisting individuals in the management of NCDs, emerging evidence points to a potential role for CHWs in health promotion and the primary prevention efforts. A review by Jeet et al. [60] provides promising evidence of the effectiveness of using CHWs to lead primary prevention interventions that address multiple risk factors at the same time (tobacco cessation, blood pressure reduction, diabetes control). Similarly, investigators in India are examining CHW-led behavior change interventions for four major NCD risk factors using brief advice in urban and rural primary care settings [61]. In Uganda, a CVD prevention program delivered by CHWs was highly acceptable among community members [62]. Furthermore, Babmoto et al. [63] found that a CHW-led intervention resulted in improvements in dietary and physical activity behaviors, as well as a 2.9× greater odds of reducing body mass index when compared to controls. This evidence suggests that CHWs should be considered a part not only of NCD treatment and control, but also in health promotion and primary prevention interventions. 

## 3. Task-Shifting: The Role of CHWs in Physical Activity Promotion

Given the growing prevalence and cost of NCDs in LMICs and the potential role of CHWs in mitigating this disease burden, it is essential that new approaches for engaging CHWs in lifestyle behavior change are developed. The SA National Department of Health’s strategic plan for NCDs addresses lifestyle behavioral change, including physical inactivity [64]. However, the planned action for this strategy is yet to be outlined. Therefore, we propose a model for the role of CHWs in promoting physical activity, which includes five key tasks, based on a larger model of service delivery developed by Collingsworth et al. [65] (Figure 1).

### 3.1. Assessment of Physical Activity

In our model, CHWs play a key role in assessing current physical activity levels of community members and patients at local health centers. Although there are multiple ways to assess an individual’s physical activity status, a single item question has been shown to be one of the most valid and reliable methods of assessing physical activity on a large scale [66]. The single question refers: “In the past week, on how many days have you done a total of 30 min or more of physical activity, which was enough to raise your breathing rate?” It can be modified to include housework and occupation, which is relevant to these communities. This question would serve as a first step in future “exercise” prescription and form the basis for further individual counseling. This brief screening may lead to additional discussion with individuals, as well as providing key information on physical activity prevalence on both an individual and community level. 

### 3.2. Conduct Basic Pre-Participation Screening and Risk Stratification

While the health benefits of physical activity are numerous, the increased risk of musculoskeletal injury, acute cardiac episodes, and medical complications cannot be overlooked [67,68]. This risk is significantly higher in older individuals and those with established cardiovascular disease risk factors or diagnosed disease. This is particularly important in the context of colliding pandemics; in an LMIC setting, physical activity risks need to be considered in the context of high rates of both communicable diseases (tuberculosis, HIV and malaria) and NCDs. Using pre-screening tests with risk stratification is essential and has been proposed by many international organizations [69]. These pre-participation screening guidelines [70,71] are designed to reduce excessive physician referrals and unnecessary barriers to physical activity participation while ensuring patient safety. They provide a pragmatic and practical approach to identifying high-risk individuals to minimize the risk of sudden cardiac death [72]. This process involves a self-assessment of risk questionnaire [70], which can be conducted by a non-physician health professional, in this case the CHWs, who would initiate referral for further medical assessment in the case of high-risk patients. Well-trained CHWs in Bangladesh and South Africa demonstrated a 96.8% level of agreement with healthcare professionals in conducting primary, non-invasive screening for cardiovascular disease risk, suggesting that task-shifting of community-based screening with CHWs along with clear strategies for implementation is a viable option [73]. We believe that our model provides a clear strategy for expanding the integration of CHWs in pre-participation screening and risk stratification within PHC. 

### 3.3. Physical Education and Counseling

Behavior change counseling, education, and advocacy at the PHC level is a feasible and cost-effective method for reducing the burden of NCDs [74] and presents a scalable opportunity for influencing population-level physical activity, if achieved through strategic and effective interventions [75]. In high-income countries, such as Sweden and New Zealand, PHC interventions led by nurses or general practitioners are generally effective in increasing physical activity levels [76,77]. However, physical activity education and counseling by nurses and medical practitioners within public healthcare clinics are not without their challenges. A lack of time, resources, and support have all been cited as barriers to effective behavioral counseling [78,79]. 

CHWs could be well situated to take over the provision or responsibility, via task-shifting, of physical activity promotion efforts. Previous studies have shown that CHWs are effective in increasing patient knowledge and stimulating behavior change in other health areas [80], benefits that could easily be extended to the physical activity domain. CHWs can play a crucial role in disseminating two key public health messages. Firstly, they can encourage low-risk individuals to participate in 150 min of moderate intensity physical activity or 75 min of vigorous intensity physical activity, or a combination of both. Secondly, everyone, irrespective of their risk profile, can be encouraged to reduce their sedentary behaviors. Indeed, the recently updated Physical Activity Guidelines [81] emphasized the importance of adopting and maintaining a habitual lifestyle of physical activity and reducing sedentary behavior—a message that CHWs are well suited and capable of providing. 

CHWs can also be empowered to provide physical activity interventions through individual counseling. As an example, CHWs can be trained on the five A’s, a model that has been evaluated for use with several behavioral change efforts [82], as well as physical activity counselling [83]. The 5 A’s, adapted for physical activity, include: (1) ***Assess***: current physical activity levels, as described above, as well as patient readiness to change, social support and self-efficacy; (2) ***Advise***: to provide a structured, individually tailored exercise prescription and message based on the findings of the first step; (3) ***Agree***: facilitate shared decision-making based on patients stage of change; (4) ***Assist***: provide support tools for maintaining desired activity change, such as provision of community exercise programs, printed material, and resources; (5) ***Arrange***: schedule a follow up visit or arrange a medical referral if needed [83].

CHWs can also be involved in community level education initiatives, as well as population-level campaigns. In low-income settings, CHWs have become the cornerstone for implementing preventive measures and upscaling of community-based interventions [84]. From a physical activity perspective, there is moderate evidence that community-wide interventions that involve large populations are effective in improving physical activity levels [85]. This is particularly true if the intervention involves intensive contact with the majority of the population over time, something that CHWs are well-placed to implement [16,22]. Community campaigns may also be particularly relevant in low-income or rural areas where access to traditional recreational physical activity resources are limited and there are greater safety concerns. 

### 3.4. Provide Basic Exercise Activities to Low-Risk Groups 

In a systematic review by Garrett et al. [86], the promotion of walking and exercise groups in addition to brief counseling was the most cost-effective methods of improving physical activity levels in a PHC setting. CHWs are more than capable of providing and supporting community-based physical activity interventions such as walking or exercise groups. In SA, Tsolekile et al. [16] found that CHWs are often already providing one-on-one and group rehabilitation exercises in conjunction with their other roles. If CHWs are trained in basic exercise prescription, they could provide or facilitate activities such as group classes, fun run/walks, and individual sessions in low-risk patients. 

### 3.5. Referral to Specialist for Individualzed Exercise in High-Risk Groups 

If the pre-screening protocol identifies a high-risk patient, those with special needs, or individuals seeking more specialized physical activity guidance, CHWs are well-positioned to manage onward referral to specialized exercise interventions led by qualified professionals. A core function of CHWs is serving as a link between the community and healthcare professionals [19], which would be no different in connecting patients to clinical exercise physiologists, biokineticists, physiotherapists, and sports medicine physicians. One (aspirational) step further would be embedding exercise professionals alongside CHWs, integrated within the health system, which would create a novel, integrated workflow with the potential to change the way PHC addresses the risk of NCDs.

## 4. Training for Success

In the re-engineering of the SA PHC system, three phases of training were outlined for CHWs working within the WBPHCOTs. Of these, the second phase is dedicated to providing CHWs with specialized training on NCDs and social support. In 2015, a national appraisal found that the training being offered was inadequate (e.g., unconducive learning spaces, poor general planning) for CHWs to finish phase 1, let alone subsequent training, prior to their community placements [44]. Challenges with insufficient training and support are encountered by CHWs in LMICs across the world [87,88,89]. However, the inclusion of NCDs as a part of phase II provides an important opportunity to incorporate physical activity training into the national curriculum. Currently, a core curriculum [20] in SA is being developed and accredited by the government to ensure that CHWs are trained to a specific standard to reinforce their role in the PHC structure. This broad curriculum includes a section on health education/promotion/communication. Although there is a section for nutrition, there is no dedicated training in pre-exercise screening and physical activity counseling. Therefore, developing a physical activity module into the curriculum is critical for improving CHWs’ knowledge and skills to effectively conduct physical activity counseling.

In developing CHW training programs, there have been calls for strengthening of the materials, delivery, and evaluation of training [57,73]. Data from Brazil suggest that 97% of CHWs desired more information on physical activity guidelines as only 3.6% understood current physical activity recommendations [90]. Furthermore, CHWs were less knowledgeable about recommending physical activity than the nurses and physicians, likely due to their training, educational backgrounds, and expectations to provide this counseling to patients. A scoping review estimated that when CHWs received physical activity training as a part of research interventions, the trainings consisted of an average of six hours [91], far less time than is typically dedicated to other health conditions, such as infectious disease and maternal and child health (e.g., both twice monthly over a year) [92,93]. Therefore, to either partially or fully shift the task of physical activity counseling to CHWs, there must be a much greater emphasis on providing regular training. Licensed professionals, such as biokineticists and sports medicine physicians, are well-positioned to assist in the training and supervision of the CHWs, providing greater standardization and accountability.

There is much variation in the length and content of CHW training models in the literature [94], and few focus on, or even include, physical activity. Opportunities for continuing education, professional recognition, empowerment, and career advancement, particularly within the field of physical activity, need to be developed to provide ongoing support [24,95]. When completed, CHWs desire formal recognition of this specialized training and their unique expertise (e.g., via credentials or certifications) to legitimize their roles. While CHWs in SA receive certification as an ancillary health care workers under the National Qualifications Framework [96], they have little other formal recognition and want clearer pathways to regular employment and professional growth [97]. In Uganda, CHWs reported that community members negatively perceived their ability to provide assistance with NCDs, believing that CHWs were only trained to address communicable diseases, sanitation, and hygiene [98]. A final important consideration is including clinic supervisors and other staff in the training to foster support and coordination around the task-shifting. Including clinic personnel increases appreciation and understanding of CHW contributions [99,100] and hosting sessions onsite provides training within the specific context and existing capacities of the health clinic [101]. 

## 5. Conclusions

The SA health care system is being redesigned to include CHWs as integral members of the community-clinical team to increase community outreach and improve patient outcomes [65]. We have presented evidence supporting the promotion of physical activity by CHWs as a regular part of their responsibilities to stem the tide of NCDs both globally and locally in SA. There is a strong case for the task-shifting of physical activity pre-participation screening, risk stratification, counseling, delivery of exercise activities, as well as onward referral to healthcare professionals [102]. Successful task-shifting requires careful organization and structuring that includes clearly defined roles, adequate and ongoing supervision and evaluation, and the delivery of high-quality training [38]. 

At the same time, this recommendation for shifting physical activity tasks to CHWs is not without its challenges. While physical activity is dubbed as a public health “best buy” for its wide reach in disease prevention [75], physical activity is often a secondary priority in the face of more immediate health concerns in LMICs. Due to scarce finances and a shortage of time, resources and personnel are often directed toward clinical activities (e.g., medical management of disease) rather than addressing preventive, lifestyle behaviors. To overcome this prioritization and resource constraints, there needs to be a high level of buy-in and support from the healthcare providers, organizational leaders, and local government to create a shift in focus toward physical activity as a viable solution for NCDs. Only with this support can such a task-shifting idea be implemented effectively and successfully. In conclusion, CHWs present a unique and exciting opportunity to safely improve physical activity levels and aid in behavioral change in LMICs. 

## Figures and Tables

**Figure 1 ijerph-20-06675-f001:**
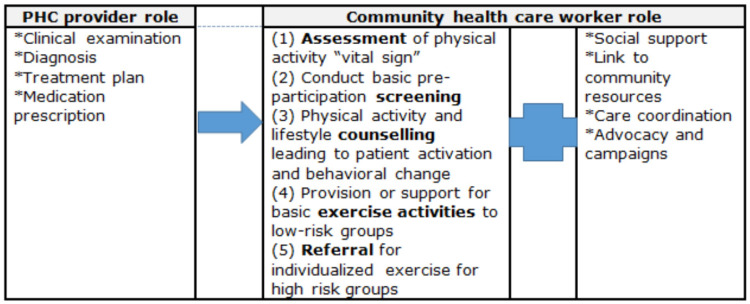
Roles of the primary health care (PHC) provider and community health care worker in the prevention and management of NCDs (adopted from Collinsworth et al. [65]; Tsolekile et al. [16]).

## Data Availability

No new data were created or analyzed in this study. Data sharing is not applicable to this article.

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
