# Peer review of "Task-Shifting: Can Community Health Workers Be Part of the Solution to an Inactive Nation?"

_ijerph, 2023, doi:10.3390/ijerph20176675_

Round 1

Reviewer 1 Report

This study is considered to be a very necessary study as it reflects the reality of South Africa.

The necessity and purpose of the study are clearly presented.

The research design that incorporates the benefits of physical activity into the medical field is judged to be very good.

Physical activity also plays a very important role in reducing the burden of medical expenses.

Researchers need additional explanations about what kind of physical education they do.

The necessity and purpose of the study were sufficiently well explained. However, it is judged that the results and discussions are relatively insufficient.

Sentences throughout the thesis were written too long. It needs to be written in short sentences so that readers can easily understand them.

Sentences throughout the thesis were written too long. It needs to be written in short sentences so that readers can easily understand them.

Author Response

Thank you for your suggestions and the opportunity to improve the paper. Point by point response is attached

Reviewer 2 Report

The article is interesting but lacks logical flow. So I would like to recommend a major revision. Detailed comments are given below;

1. Revise your abstract and remove practical implications from the abstract

2. Authors should reorganize the content of the whole article with scientific facts and figures. It could make the article more interesting for readers.

3. “The burden of non-communicable diseases” needs more latest and more comprehensive explanation.  And also explain how the inactivity problem can be solved through physical activity.

4. Figure 1 is used by the authors in the article taken from an article. So they should have to mention “figure is adopted.”

5. Conclusion section doesn’t conclude the main theme of the study. The conclusion should be comprehensive. The practical implications section should be separated with a heading and then written in details

6. Future scope of the study is an important aspect which I’m unable to find  

English editing is required to improve the quality

Author Response

Thank you for your comments and the opportunity to improve the paper. A point by point response is attached

Round 2

Reviewer 1 Report

The researcher is judged to have made good corrections based on the contents of the review document.

Researchers put a lot of effort into fixing many parts.

Reviewer 2 Report

NA